# D-2-Hydroxyglutarate and L-2-Hydroxyglutarate Inhibit IL-12 Secretion by Human Monocyte-Derived Dendritic Cells

**DOI:** 10.3390/ijms20030742

**Published:** 2019-02-10

**Authors:** Ines Ugele, Zugey Elizabeth Cárdenas-Conejo, Kathrin Hammon, Monika Wehrstein, Christina Bruss, Katrin Peter, Katrin Singer, Eva Gottfried, Jakob Boesch, Peter Oefner, Katja Dettmer, Kathrin Renner, Marina Kreutz

**Affiliations:** 1Ear, Nose and Throat Department, University Hospital Regensburg, Regensburg 93053, Germany; ines.ugele@ukr.de; 2Department of Internal Medicine III, University Hospital Regensburg, Regensburg 93053, Germany; elizabeth_cardenas_86@outlook.com (Z.E.C.-C.); Kathrin.Hammon@ukr.de (K.H.); monika.wehrstein@ukr.de (M.W.); christina.bruss@ukr.de (C.B.); katrin.peter@ukr.de (K.P.); katrin.singer@ukr.de (K.S.); eva.gottfried@gmx.de (E.G.); jakobboesch@icloud.com (J.B.); Kathrin.Renner-Sattler@klinik.uni-regensburg.de (K.R.); 3Institute of Functional Genomics, University of Regensburg, Regensburg 93053, Germany; Peter.Oefner@klinik.uni-regensburg.de (P.O.); Katja.Dettmer@klinik.uni-regensburg.de (K.D.); 4Regensburg Center for Interventional Immunology, Regensburg, Regensburg 93053, Germany

**Keywords:** isocitrate dehydrogenase, hydroxyglutarate, dendritic cells, tumor environment, activation

## Abstract

Mutations in isocitrate dehydrogenase (IDH) or a reduced expression of L-2-hydroxyglutarate (HG)-dehydrogenase result in accumulation of D-2-HG or L-2-HG, respectively, in tumor tissues. D-2-HG and L-2-HG have been shown to affect T-cell differentiation and activation; however, effects on human myeloid cells have not been investigated so far. In this study we analyzed the impact of D-2-HG and L-2-HG on activation and maturation of human monocyte-derived dendritic cells (DCs). 2-HG was taken up by DCs and had no impact on cell viability but diminished CD83 expression after Lipopolysaccharides (LPS) stimulation. Furthermore, D-2-HG and L-2-HG significantly reduced IL-12 secretion but had no impact on other cytokines such as IL-6, IL-10 or TNF. Gene expression analyses of the IL-12 subunits p35/IL-12A and p40/IL-12B in DCs revealed decreased expression of both subunits. Signaling pathways involved in LPS-induced cytokine expression (NFkB, Akt, p38) were not altered by D-2-HG. However, 2-HG reprogrammed LPS-induced metabolic changes in DCs and increased oxygen consumption. Addition of the ATP synthase inhibitor oligomycin to DC cultures increased IL-12 secretion and was able to partially revert the effect of 2-HG. Our data show that both enantiomers of 2-HG can limit activation of DCs in the tumor environment.

## 1. Introduction

Point mutations in IDH1 and IDH2 are detected in different types of malignancies such as glioma and glioblastoma [1,2,3], acute myeloid leukemia (AML), and head and neck squamous cell carcinoma (HNSCC). Data from glioma patients show a positive correlation between mutated IDH and overall survival [4], whereas the prognostic impact of IDH mutations in AML varies across studies [5]. All mutations are generally heterozygous and it has been shown that mutated IDH gains the ability to convert αKG into the oncometabolite D-2-HG.

Moreover, increased levels of 2-HG in tumor tissues have been described in tumor entities lacking IDH mutations. In breast cancer [6], 2-HG accumulation is based on increased glutamine uptake and glutaminolysis. In renal cell carcinoma [7], the L-enantiomer of 2-HG (L-2-HG) accumulates based on reduced expression of L-2-HG dehydrogenase, the enzyme which normally degrades L-2-HG. In both cancer entities, 2-HG accumulation is associated with worse prognosis.

T-cells and DCs play a pivotal role in anti-tumor immune response. Accordingly, good prognosis in cancer patients is associated with high T-cell density [8]. T-cell infiltration has also been shown to have a positive impact on survival probability in patients with malignant gliomas [9,10]. Furthermore, T-cell associated genes are enriched in high-grade astrocytoma in long-term compared with short-term survivors [11]. Interestingly, in renal cell carcinoma as well as in head and neck cancers, T-cell infiltration seems to be controlled by the metabolic phenotype of the tumor, as an accelerated tumor glucose metabolism has been associated with a low CD8 effector T-cell infiltration [12].

Several studies suggest an interplay between tumor metabolism and immune cell metabolism [13]. The accelerated glucose and amino acid uptake of cancer cells may limit nutrient availability for tumor-infiltrating immune cells. Furthermore, metabolic waste products, such as lactate or 2-HG, have been shown to alter immune cell function. Human monocyte activation and DC differentiation is compromised by lactic acid [14,15] and T- and NK-cell function is impaired in lactate-rich tumors [16]. Recently, Bottcher et al. reported that D-2-HG is taken up by T-cells and is enriched in T-cells in patients with AML blasts that harbor mutant IDH [17]. They showed that exogenous D-2-HG shifts the metabolism of activated human T-cells from aerobic glycolysis towards respiration, thereby favoring the differentiation of regulatory T cells but diminishing Th17 polarization [18]. In line, IDH mutations are associated with low CD8^+^ T-cell abundance in gliomas, indicating an impact of 2-HG on T-cells not only in vitro but also in tumor patients [19,20]. In line, Bunse et al. recently described that inhibition of the enzymatic function of mutant IDH1 improves antitumor immunity in IDH1-mutant tumors [21]. Moreover, IDH mutant gliomas downregulate the NKG2D gene via epigenetic silencing, which in turn reduces NK-cell activation [22]. These data demonstrate a possible involvement of 2-HG in the immune evasion of tumors.

The influence of D-2-HG on human myeloid cells, especially DCs, has not been investigated so far. DCs are specialized antigen presenting cells that acquire, process, and present tumor-associated antigens to T-cells for the induction of a tumor-specific immune response [23]. Mature and functional active DCs seem to be absent in many tumor entities, resulting in a failure to generate a tumor-specific T-cell response; possible underlying mechanisms include the accumulation of immunosuppressive factors. Cytokines such as IL-10, TGF-β, and vascular endothelial growth factors are known to reduce DC development and function in the tumor environment [24,25]. DC activation depends on the switch from a mitochondrial to a glycolytic metabolism which supports the migratory ability of DCs [26]. Based on the obvious importance of DCs in the anti-tumor immune response, we investigated the effect of D-2-HG and L-2-HG on activation and maturation of human DCs in vitro. Our data underline the role of D-2-HG as an important, immunosuppressive metabolite in the tumor environment.

## 2. Results

### 2.1. Uptake of D-2-HG by DCs Has No Impact on DC Viability and Maturation

It has been shown that human T-cells take up 2-HG, which is accompanied by reprogramming of T-cell metabolism and function. Here we measured cellular uptake of D-2-HG and L-2-HG by monocyte-derived human DCs. Immature DC were stimulated with 100 ng/mL Lipopolysaccharides (LPS) and treated with 10 mM D-2-HG or L-2-HG for 1 h or 24 h, respectively. Viability was measured using the CASY system and revealed no impact of D-2-HG or L-2-HG on the viability of DCs (Figure 1a). Intracellular levels of D-2-HG were analyzed by mass spectrometry and detected endogenous 2-HG in DCs without 2-HG supplementation. Under treatment with 10 mM D-2-HG or L-2-HG, intracellular levels of 2-HG in DCs increased about 100-fold (Figure 1b). This result demonstrates the capacity of DCs to take up 2-HG. Next, we analyzed the impact of D-2-HG on the maturation of DCs and determined the expression of typical maturation markers after 24 h incubation with LPS by flow cytometry. 2-HG neither changed classical DC marker expression such as CD1a nor co-stimulatory molecule expression such as CD80 or CD86. However, the upregulation of CD83 was significantly decreased (Figure 1c,d). Similar results were obtained using the L-enantiomer. In the absence of LPS, both enantiomers did not alter surface marker expression (Figure 1e).

### 2.2. D-2-HG Reduces IL-12 Secretion and mRNA Expression in DCs

Beside expression of costimulatory molecules, secretion of IL-12 by DCs is an important signal for the induction of T-cell differentiation. Hence, we analyzed the capacity for IL-12 production by DCs after treatment with D-2-HG and its enantiomer L-2-HG. Supernatants of DCs were collected and cytokine production was determined by ELISA (Figure 2). D-2-HG as well as L-2-HG significantly reduced IL-12 p70 secretion in a dose-dependent manner, indicating that not only D-2-HG but also its physiological enantiomer has suppressive activity (Figure 2a,b). Based on these results, we were interested in whether the production of other cytokines such as IL-10, IL-6, or TNF is also affected by D-2-HG or L-2-HG. D-2-HG as well as L-2-HG slightly upregulated IL-10 secretion, whereas IL-6 and TNF secretion were downregulated by trend (Figure 2c–e). As IL-12 is important for T-cell stimulation, we next determined the capacity of DCs to stimulate T-cells in a mixed lymphocyte reaction. Neither D-2-HG nor L-2-HG had an impact on T-cell proliferation (Figure 2f).

To confirm the reduction of IL-12 p70 secretion, we analyzed intracellular levels of the IL-12 subunits p35 and p40 by flow cytometry. For this purpose, DCs were stimulated with LPS in the presence or absence of D-2-HG or L-2-HG and a protein transport inhibitor preventing the release of IL-12. The percentage of double IL12 p35 and p40 positive cells were reduced in LPS plus D-2-HG treated DCs compared to LPS-stimulated control DCs (Figure 3a).

Next, we analyzed gene expression levels of the IL-12 subunits p35 (IL-12A) and p40 (IL-12B). LPS-stimulated DCs were treated with D-2-HG or L-2-HG for 4 or 24 h and gene expression was determined by RT-qPCR analysis. After stimulation with 100 ng/mL LPS, we detected a strong expression of IL-12A and IL-12B genes. Treatment with D-2-HG or L-2-HG significantly decreased expression after 4 h (Figure 3b,c); after 24 h the effect was no longer significant for IL-12A but was still significant for the expression of IL-12B (Figure 3d,e).

### 2.3. D-2-HG and L-2-HG Have No Impact on LPS-Associated Signaling Pathways in DCs

As we detected a strong impact of 2-HG on IL-12 secretion and gene expression, we investigated signaling pathways associated with LPS-induced cytokine production. As exposure of DCs to LPS is known to activate nuclear factor (NF)-kB, Akt, and p38 pathways [27] we stimulated DCs with LPS and 2-HG and analyzed the expression of IkBα, P-Akt, and P-p38 by western blot. Proteasomal degradation of IkBα is a key step in the regulation of the NFκB pathway and leads to nuclear translocation and activation of gene expression. As expected, LPS reduced IkBα expression. However, 2-HG treatment showed no significant effect on IκBα expression (Figure 4a).

Akt is a serine/threonine-specific protein kinase and is known to directly regulate IL-12 production [28]. LPS increased the amount of P-Akt and reduced the Akt content at the same time, although statistical significance was not reached. D-2-HG and L-2-HG had no impact on these alterations (Figure 4b). Next, we analyzed the effect of D-2-HG and L-2-HG on the p38 signaling pathway. LPS stimulation had no significant impact on the expression of P-p38, and addition of 2-HG did not change P-p38 levels (Figure 4c). These results suggest that suppression of IL-12 production is not related to an impairment of the LPS-associated NF-kB, Akt, and p38 signaling pathways, although other pathways might be involved.

### 2.4. 2-HG Reprograms DC Metabolism—Impact of Mitochondrial Activity on IL-12 Secretion

It has been shown that stimulation and maturation of murine and human DCs with LPS stimulates glycolysis and decreases respiration [29,30]. In order to evaluate the effect of 2-HG on glycolysis and respiration we measured lactate levels in cell culture supernatants and utilized high-resolution respirometry to determine respiration. LPS increased lactate secretion which was partially suppressed by the addition of 2-HG (Figure 5a). In addition, LPS stimulation significantly decreased oxygen consumption of DCs. This effect was reduced in the presence of 2-HG (Figure 5b,c).

Oxygen consumption related to ATP production was also significantly reduced after LPS addition and this effect was prevented by the addition of D-2-HG. Furthermore, the capacity of the electron transfer system was higher in D-2-HG treated cells (data not shown). These data point towards reduced substrate flux into mitochondria upon stimulation with LPS, which was counteracted by D-2-HG. These results suggest that LPS stimulation shifts the metabolism of DCs from Oxidative phosphorylation (OXPHOS) to aerobic glycolysis. D-2-HG prevents these metabolic alterations, which might contribute to a reduced IL-12 secretion. To analyze whether the reduction in mitochondrial respiration is directly linked to IL-12 production, we inhibited mitochondrial activity by oligomycin and measured IL-12 secretion in DC supernatants. IL-12 production was positively affected by treatment with low concentrations (0.05 µM) of oligomycin (Figure 5d). A combination of D-2-HG and L-2-HG with low concentrations of oligomycin partially rescued IL-12 secretion (Figure 5e).

## 3. Discussion

In different tumor entities 2-HG accumulation is associated with worse prognosis and recent data suggest a negative correlation between 2-HG levels in gliomas and the anti-tumor immune response [21,31]. The impact of D-2-HG on T-cells has been documented recently [17,21] but no data are available regarding the effects of 2-HG on human DCs. Up to now, most studies have concentrated on the effects of D-2-HG as IDH mutations are associated with elevated levels of the D-enantiomer. However, in kidney tumors elevations of the L-enantiomer have been described [7]. Here, accumulation of L-2-HG is mediated by a reduced expression of L-2-HG-dehydrogenase, which metabolizes L-2-HG back to ketoglutarate. Therefore, not only D-2-HG but also L-2-HG may act as an oncometabolite. Interestingly, Tyrakis et al. have described the suppression of IFNα production, decreased cytotoxicity, and reduced viability after treatment of murine T-cells with 0.5 mM L-2-HG, which represents a very low concentration compared to 20 mM 2-HG levels described in glioma [21,32]. In our experiments, L-2-HG and D-2-HG had comparable effects on DCs, indicating that not only D-2-HG but also L-2-HG is a potent modulator of the immune response.

IL-12 p70 secretion by DCs was significantly decreased in LPS-stimulated DCs in the presence of physiological amounts of D-2-HG or L-2-HG (10 mM). In contrast, secretion of other cytokines was not significantly changed. In line, the transcription of IL-12A and IL-12B was decreased. As IL-12 is not only a marker for DC activation but also a marker for successful maturation, we analyzed the impacts of D-2-HG and L-2-HG on DC maturation. After seven days of differentiation with GM-CSF and IL-4, immature DCs were incubated for another 24 h with LPS. With the exception of CD83 expression, no changes in surface marker expression were detected, indicating a very specific effect of 2-HG on DC maturation.

Based on these data, we investigated the impact of 2-HG on pathways involved in LPS signal transduction. Upon LPS stimulation, DCs activate NFκB, Akt, and p38 pathways [27]. Activation of the PI3K/Akt pathway has been shown to inhibit IL-12 secretion in human monocytes [28] and in murine DCs [33]; however, the opposite effect has been described for human monocyte-derived macrophages and DCs [34]. Mitogen-activated p38 protein kinases are also involved in the regulation of LPS-stimulated IL-12 gene expression in macrophages [35]. In our experiments, all studied pathways were not significantly changed by 2-HG treatment. Therefore, these pathways cannot be responsible for the negative impact on IL-12 production.

It is well-known that metabolism is important for the activation, differentiation, and function of different types of immune cells. DCs increase glucose metabolism and decrease OXPHOS after LPS stimulation [26]. Therefore, we wondered whether inhibition of IL-12 secretion by 2-HG could be associated with an altered metabolism. A metabolic shift from OXPHOS towards glycolysis is regarded as an important event in the activation process of LPS-stimulated macrophages [36] and monocytes [14]. Furthermore, DC maturation has been linked to a reduction in OXPHOS and an increase in glycolysis [30]. In our experiments, LPS treatment changed the metabolic profile of DCs in a similar manner: LPS diminished OXPHOS in DCs, whereas lactate secretion was increased. 2-HG treatment supported mitochondrial activity and slightly diminished glycolysis. In line, we could show that 2-HG triggers an increase in OXPHOS and reduces the glycolytic activity of primary human T-cells [17]. IDH1-mutated human oligodendroglioma cells with endogenous 2-HG production also displayed increased OXPHOS activity in a xenograft mouse model [37]. Similar results have been obtained by Grassian et al. [38] showing that tumor cells expressing mutant IDH1 are more sensitive to pharmacologic inhibition of oxidative metabolism compared to IDH1 wildtype cells. Other studies have shown contradictory effects of 2-HG on respiration in tumor cells. Chan and colleagues have observed that D-2-HG inhibits mitochondrial cytochrome c oxidase (COX) activity in AML cells [39]. Fu et al. have reported an inhibition of respiration and ATP synthase in glioblastoma cells by D-2-HG and L-2-HG [40]. These data indicate that primary cells and tumor cells may differ in their metabolic response to 2-HG. Furthermore, exogenous application of 2-HG versus endogenous production in IDH mutated cells may explain the different effects on cell metabolism.

Malinarich has stated that tolerogenic DCs are characterized by increased OXPHOS, reduced glycolysis, and low IL-12 production [30]. This phenotype resembles DCs treated with 2-HG, and could indicate that 2-HG induces tolerogenic DCs. However, as costimulatory molecules such as CD80 and CD86 as well as MHC II molecules were not altered by 2-HG treatment, it seems unlikely that HG-treated DCs cannot induce T-cell proliferation. In line, we did not find an impact of 2-HG on antigen presentation as T-cell proliferation was not impaired. However, these data do not exclude the possibility that HG-treated DCs may alter IFN-γ secretion by T-cells. It thus appears that different aspects of LPS-induced DC maturation are regulated by different pathways and 2-HG targets mainly metabolic changes and therefor IL-12 production. Further studies should clarify the impact of 2-HG treatment on antigen presentation by DCs.

Interestingly, blocking mitochondrial respiration leads to an increase of IL-12 levels in the presence of 2-HG. As drugs targeting respiration, such as metformin, are already being tested in tumor patients, it may be possible to block OXPHOS in patients with IDH mutations or reduced L-2-HG-dehydrogenase activity to relieve immunosuppression.

In conclusion, our results indicate that both the oncometabolite D-2-HG but also its enantiomer L-2-HG impairs DC function by downregulation of IL-12 secretion, which potentially may contribute to the immune escape of tumor cells in a tumor environment with 2-HG accumulation.

## 4. Materials and Methods

### 4.1. Cell Isolation and Culture

Peripheral blood mononuclear cells (PBMCs) of healthy donors were isolated by leukapheresis. Leukapheresis was approved by the Ethics committee of the University Hospital Regensburg (Ethic Vote July 2010 #09/066b and #09/066c); all human participants gave written informed consent. Human monocytes were isolated from PBMCs by density gradient centrifugation over Ficoll/Hypaque followed by counterflow centrifugation elutriation [41]. Immature dendritic cells (iDC) were generated from human blood monoctes. Monocytes were cultured in culture flasks at a concentration of 1 × 10^6^ cells/1.5 mL in RPMI 1640 (Gibco, Waltham, MA, USA) for seven days. The medium was supplemented with 10% fetal calf serum (FCS), 2 mM L-glutamine (Biochrom, Berlin, Germany), 50 U/mL penicillin (Gibco, Waltham, MA, USA), 50 U/mL of streptomycin (Gibco), 225 U/mL granulocyte macrophage colony stimulating factor (GMCSF, Peprotech, Hamburg, Germany), and 144 U/mL recombinant IL-4 (Peprotech). Cell number, cell size, and cell viability were determined using the CASY cell analyzer system (Casy^®^ Modell TT, OLS Omni Life Science, Bremen, Germany). Appropriate cursor settings for determining cell number and viability were established for each cell type.

### 4.2. Mixed Lymphocyte Reaction (MLR)

Immature dendritic cells were generated from human blood monocytes. Monocytes were cultured in culture flasks at a concentration of 1 × 10^6^ cells/1.5 mL in RPMI 1640 for five days with 10% fetal calf serum (FCS), 225 U/mL granulocyte macrophage colony stimulating factor (GMCSF, Peprotech) and 144 U/mL recombinant IL-4 (Peprotech). LPS (Enzo, Life Sciences, Farmingdale, NY, USA) and D-2-HG or L-2-HG (both Sigma-Aldrich, St. Louise, MO, USA) were added to iDC on day five in the culture flasks. For MLR, different amounts of monocyte-derived DC were harvested on day seven. DCs were washed and cocultured with allogeneic human T lymphocytes in RPMI containing 5% AB serum, L-glutamine (2 mmol/L), penicillin (50 U/mL), and streptomycin (50 mg/mL). On day five of coculture, 0.5 µCi/0.2 mL [^3^H]-thymidine (Hartmann Analytic, Braunschweig, Germany) was added, and incorporated radioactivity was quantified after 24 h by means of a beta counter (Perkin Elmer, Gaithersburg, Waltham, MA, USA). All samples were analyzed in triplicate.

### 4.3. ELISA

For the determination of extracellular cytokine concentrations, supernatants were harvested and cytokines were determined by means of commercially available ELISA kits (Duoset ELISA, R&D Systems, Minneapolis, MN, USA).

### 4.4. Flow Cytometry

To analyze DC surface markers, monocytes were cultured in RPMI medium supplemented with IL-4 and GM-CSF for seven days. DCs were treated with 100 ng/mL LPS for 24 h in the absence or presence of 10 mM D-2-HG or L-2-HG. Cells were harvested and stained with anti-CD1a & anti-HLA-DR (Beckman Coulter, Krefeld, Germany), anti-CD80 (Biolegend, San Diego, CA, USA), anti-CD83 (eBioscience, San Diego, CA, USA), and anti-CD86 (BD Bioscience, Franklin Lakes, NY, USA) at 4 °C for 30 min. After washing, DCs were resuspended in FACS wash buffer. Flow cytometric measurement was performed using a BD FACS Calibur instrument (BD Bioscience).

For intracellular detection of IL-12 p35 and IL-12 p40, monocytes were cultured in RPMI medium supplemented with IL-4 and GM-CSF. After seven days of culture, cells were treated with 100 ng/mL LPS with or without 10 mM D-2-HG in the presence of a protein transport inhibitor containing Monensin (BD GolgiStop^TM^, BD Bioscience, Franklin Lakes, NY, USA) for 16 h. DCs were washed, permeabilized, and fixed using the BD Cytofix/Cytoperm^TM^ Kit (BD Biosciences), followed by staining with anti-IL-12 p40 (R&D), anti-IL-12 p35 (R&D), and the respective isotype controls. Cells were analyzed using a BD FACS Calibur instrument (BD Bioscience).

### 4.5. Determination of 2-HG Uptake by Liquid Chromatography-Mass Spectrometry

2.5 × 10^6^ DCs/well were seeded in a 6-well plate in a final volume of 4 mL and stimulated with 100 ng/mL LPS in the absence or presence of 10 mM D-2-HG or L-2-HG for 24 h. Cell pellets were precipitated in 500 µL methanol (80%) supplemented with 10 µL of stable isotope-labeled internal standard solution (2,3,3-d3-2HG, C/D/N Isotopes Inc., Pointe-Claire, QC, Canada; 100 μM in water). The pellet was washed twice consecutively with 100 µL methanol (80%). Further sample preparation and HPLC-MS/MS analysis was performed as described in Voelxen et al. [42]. Metabolite concentrations were normalized to cell counts.

### 4.6. Quantitative Real-Time PCR (qPCR)

2.5 × 10^6^ DCs/well were seeded in a 6-well plate in a final volume of 4 mL and stimulated with 100 ng/mL LPS in the absence or presence of 10 mM D-2-HG or L-2-HG for 4 or 24 h. Total RNA of monocytes or macrophages was obtained using the RNeasy Mini Kit (Quiagen, Hilden, Germany). Complementary DNA (cDNA) was synthesized with a M-MLV Reverse Transcriptase kit (Promega, Madison, WI, USA) and amplified by qPCR with the QuantiFastSYBR Green PCR Kit (Quiagen) using the Mastercycler Ep Realplex (Eppendorf, Hamburg, Germany). Primer sequences were purchased from Eurofins MWG Operon, Germany. All data were normalized to 18S cDNA RNA. PCR reaction was carried out in a 96-well plate format adapted to the Eppendorf Realplex Mastercycler EpGradient S system. The amount of amplified DNA relative to reference gene 18S rRNA was measured through the emission of light by SYBR green dye after each extension step. A melting curve was monitored to determine the specificity of the amplification product. Primer sequences—IL-12A: Sense: GAAGATGTACCAGGTGGAGTTCAAGAC, Antisense: GCTCATCACTCTATCAATAGTCACTGCC; IL-12B: Sense: ACCAGCAGCTTCTTCATCAGGGAC, Antisense: ACGCAGAATGTCAGGGAGAAGTAGGA.

### 4.7. Western Blot Analysis

2.5 × 10^6^ DCs/well were seeded in a 6-well plate in a final volume of 4 mL and stimulated with 100 ng/mL LPS in the absence or presence of 10 mM D-2-HG for 1 or 24 h. Phospho-proteins were extracted as described earlier [43] and separated on a denaturating 12% acrylamide gel. After western blotting, membranes were stained with anti-ikB-β (Santa Cruz, Dallas, TX, USA), anti-Akt (Cell Signaling Technology, Danvers, MA, USA), anti-P-Akt (Cell Signaling Technology), anti-p38 (Cell Signaling Technology), anti-P-p38 (Cell Signaling Technology), or anti-β-actin (Sigma Aldrich), in dry milk (5%), and detection was performed by chemiluminescence (ECL). Actin expression was used as a loading control (Sigma Aldrich, St Louis, MO, USA). Densitometric analyses were performed by means of the ChemiDoc MP Imaging System and Image LabTM software (Bio-Rad Laboratories, Hercules, CA, USA).

### 4.8. High-Resolution Respirometry

Mitochondrial respiratory activity was determined by high-resolution respirometry using the oxygraph O2-k (Oroboros Instruments, Innsbruck, Austria). DCs were resuspended in fresh culture medium and respiration of 1 × 10^6^ cells/mL was measured at 37 °C. Data acquisition and analysis was performed with DatLab4 software (Oroboros, Innsbruck, Austria), including calculation of the time derivative of oxygen concentration, signal deconvolution dependent on the response time of the oxygen sensor, and correction for instrumental background oxygen flux [44]. Data points were recorded at 1 s time intervals. Routine respiration was measured in the first 20 min, after which D-2-HG or medium was added. When determining the effect of D-2-HG on DCs, LPS was added 10 min after D-2-HG incubation. Respiration was measured for one hour, leak respiration was determined after addition of oligomycin (2 μg/10^6^ cells), capacity of complex I to IV (ETS) after a stepwise titration of FCCP (final concentration of 3 μM, 1 μM by step), and residual oxygen consumption (ROX) not related to the respiratory system was measured after addition of rotenone (0.5 μM) and myxothiazol (2.5 μM). All respiratory parameters were corrected for ROX, and mitochondrial oxygen consumption related to ATP production was calculated as the difference between ROUTINE and LEAK respiration.

### 4.9. Statistics

Statistics were calculated using Graphpad Prism, Version 6 (La Jolla, CA, USA). Comparisons between groups were performed using the appropriate statistical methods depending on Gaussian distributions and number of groups and variables.

## Figures and Tables

**Figure 1 ijms-20-00742-f001:**
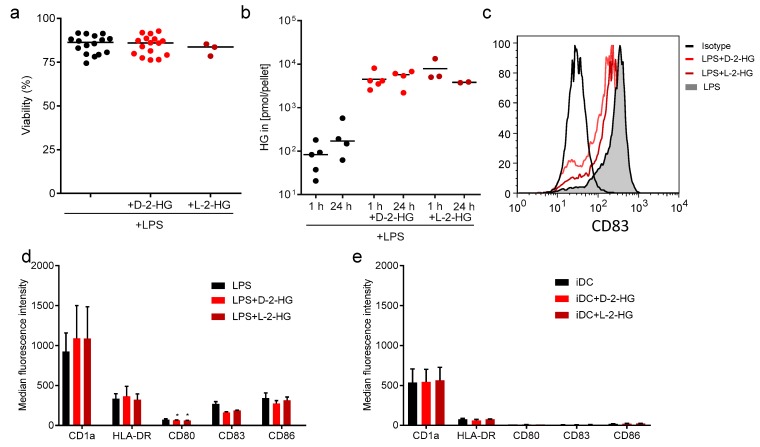
Uptake of 2-hydroxyglutarate (HG) by dendritic cells (DCs) has no impact on DC viability and maturation. (**a**) Human DCs were differentiated from elutration-separated monocytes in the presence of FCS, GM-CSF, and IL-4. On day seven, immature DCs were stimulated with 100 ng/mL LPS in the presence or absence of 10 mM 2-HG for 24 h and viability was analyzed using the CASY cell analyzer system. Data represent the median of 10 independent experiments with DCs from different donors. (**b**) Uptake of D-2-HG and L-2-HG by DC was analyzed after stimulation with 100 ng/mL LPS in the presence or absence of 10 mM 2-HG for 1 h or 24 h. Cells were washed with PBS and intracellular levels of 2-HG were analyzed by mass spectrometry. Statistical analysis was performed with the Wilcoxon-Test (** *p* ≤ 0.01). (**c**,**d**) The expression of typical DC markers on DCs stimulated for 24 h with 100 ng/mL LPS in the absence or presence of 2-HG was analyzed by flow cytometry. (**e**) As a control, DCs were cultured in the absence or presence of 2-HG without LPS stimulation.

**Figure 2 ijms-20-00742-f002:**
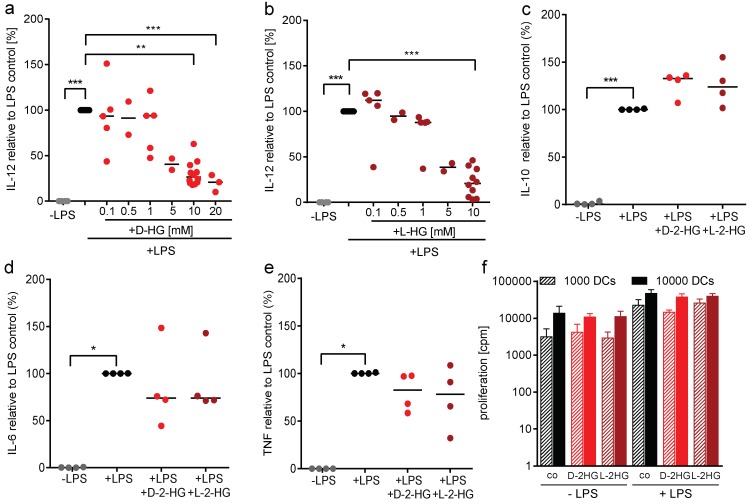
2-HG modulates LPS-stimulated cytokine secretion by DCs. Monocyte-derived DCs were plated in 6-well plates. DCs were activated with 100 ng/mL LPS and in parallel exposed to D-2-HG or L-2-HG for 24 h. Supernatants were collected and measured for IL-12 (**a**,**b**), IL-10 (**c**), IL-6 (**d**), and TNF (**e**). The measurements were performed by commercially available ELISA’s. Data represent the median of four independent experiments. Statistical analysis was tested using Dunnet’s multiple comparison test (* *p* ≤ 0.05, ** *p* ≤ 0.01, *** *p* ≤ 0.001). (**f**) Antigen presentation was analyzed in a mixed lymphocyte reaction. DCs were cultured in flasks in the absence or presence of LPS and 2-HG for 24 h, harvested, and cocultured with allogeneic T cells for another five days. Proliferation was determined by [^3^H]-thymidine incorporation.

**Figure 3 ijms-20-00742-f003:**
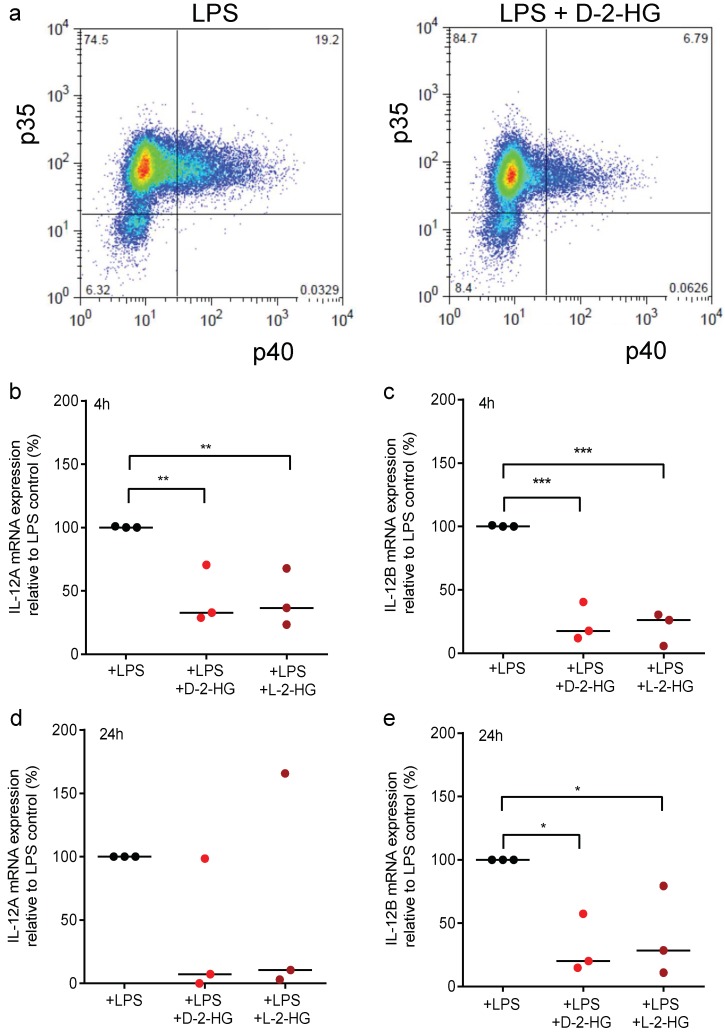
D-2-HG reduces intracellular IL-12 levels and mRNA expression in DCs. (**a**) Monocyte-derived immature dendritic cells (iDCs) were activated with 100 ng/mL LPS with or without D-2-HG in the presence of monensin. Intracellular levels of IL-12 p35 and IL-12 p40 were analyzed by flow cytometry. One representative experiment out of three is shown. (**b**,**c**) Monocyte-derived iDCs were activated with 100 ng/mL LPS with or without D-2-HG or L-2-HG for 4 h (**b**,**c**) or 24 h (**d**,**e**). RNA was isolated from cell lysates. After reverse transcription, the samples were analyzed by RT-qPCR. Gene expression of IL-12p70 subunits IL-12A and IL-12B relative to 18S RNA are shown. Data represent the median of three independent experiments. Statistical analysis was tested using an Ordinary One-Way Test with Dunnet’s multiple comparison test (* *p* ≤ 0.05, ** *p* ≤ 0.01, *** *p* ≤ 0.001).

**Figure 4 ijms-20-00742-f004:**
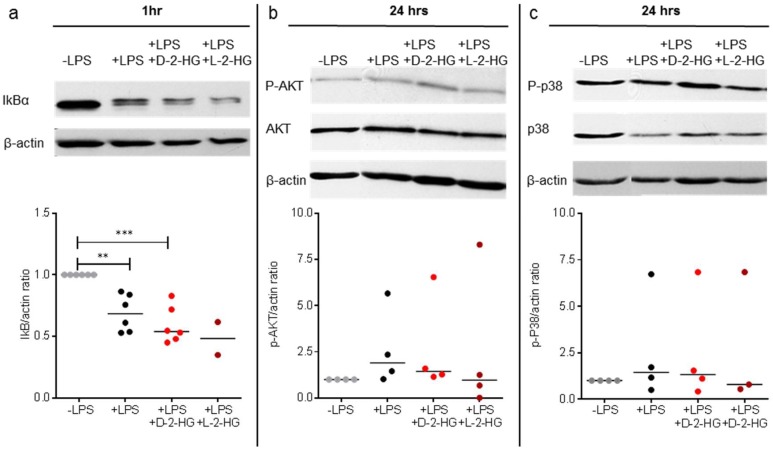
2-HG has no impact on LPS-associated signaling. DCs were stimulated with LPS (100 ng/mL) and treated with or without 10 mM 2-HG. After 1 h (IκB) or 24 h (P-Akt, P-p38) of treatment, cell pellets were lysed and analyzed by western blot. (**a**) IκB expression was analyzed in DCs treated with LPS alone or in combination with 2-HG. Data represent the median of six independent experiments. Statistical analysis was tested with the Kruskal-Wallis Test (* *p* ≤ 0.01, *** *p* ≤ 0.001). (**b**,**c**) P-Akt and P-p38 protein expression in DCs treated with LPS alone or in combination with 2-HG were analyzed. Data represent the median of four independent experiments. Statistical analysis was performed using the Kruskal-Wallis test.

**Figure 5 ijms-20-00742-f005:**
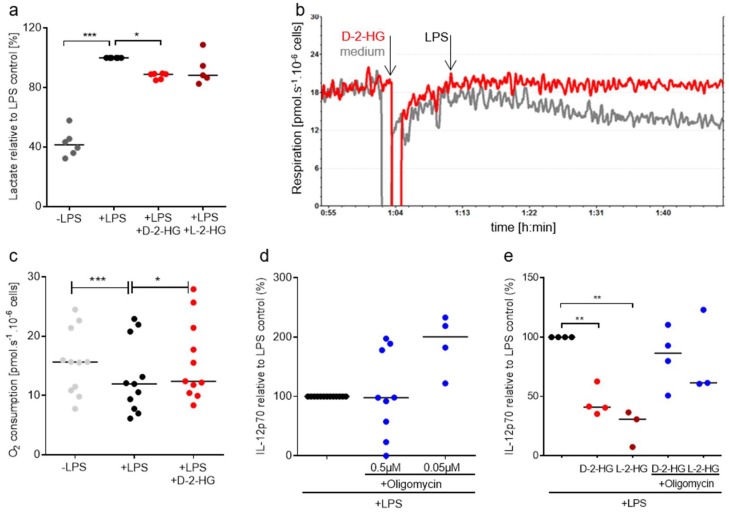
2-HG reprograms DC metabolism and thereby reduces IL-12 secretion. (**a**) Lactate secretion was determined in supernatants of DCs incubated in the presence or absence of 2-HG. (**b**,**c**) Monocyte-derived DCs were placed in oxygraph chambers in culture medium. After stabilization of respiration, 2-HG or medium was added to the chambers. After 10 min, cells were activated with 100 ng/mL LPS and oxygen consumption was monitored for one hour. (**b**) One representative experiment out of eight is shown. (**c**) Data represent the median of 11 independent experiments. Statistical testing was performed using a RM One-Way ANOVA test with Holm Sidak’s multiple comparison test (* *p* ≤ 0.05). (**d**) To evaluate the effect of oligomycin on IL-12 production by DCs, 0.2 × 10^6^ monocyte-derived DCs were plated in 24-well plates and treated with 100 ng/mL LPS and the indicated concentrations of oligomycin. IL-12 was measured in supernatants by commercially available ELISA. Data represent the median of at least four independent experiments. Statistical significance was tested using the Kruskal Wallis test. (* *p* ≤ 0.05). (**e**) Monocyte-derived DCs were plated in 24-well plates and treated with 100 ng/mL LPS and with 2-HG in combination with oligomycin. IL-12 was measured in supernatants by commercially available ELISA. Data represent the median of at least four independent experiments. Statistical significance was tested using the Kruskal Wallis test. (* *p* ≤ 0.05, ** *p* ≤ 0.01, *** *p* ≤ 0.001).

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
