# Peer review of "D-2-Hydroxyglutarate and L-2-Hydroxyglutarate Inhibit IL-12 Secretion by Human Monocyte-Derived Dendritic Cells"

_ijms, 2019, doi:10.3390/ijms20030742_

Round 1

Reviewer 1 Report

This manuscript is well written and shows interesting finding in DC biology, the data could have significant influence on tumor therapy .

Some minor changes suggested:

add one representative flow data of CD83 expression on DC treated with D-2-HG and L-2-HG in Fig 1.

change the sentence at line 159: is not related to LPS-associated  NF-KB, AKT or p38 activation pathways although other signaling might be involved.

Correction the statistic analysis error in Fig 5, line 175 **** p<0.0001 , ns not shown in the figure 5.

add the source of D-2-HG and L-2-HG ( company??) in line 280. 

Author Response

add one representative flow data of CD83 expression on DC treated with D-2-HG and L-2-HG in Fig 1.

-We included one representative experiment (Figure 1c).

change the sentence at line 159: is not related to LPS-associated  NF-KB, AKT or p38 activation pathways although other signaling might be involved.

-We changed the sentence according to your suggestion. 

Correction the statistic analysis error in Fig 5, line 175 **** p<0.0001 , ns not shown in the figure 5.

-We corrected our mistake. 

add the source of D-2-HG and L-2-HG ( company??) in line 280. 

-We incuded the company.

Reviewer 2 Report

General comment

The authors examined the effects of D-2-hydroxyglutamate (HG) and L-2-HG on dendritic cell phenotypes and functions. The aim of the study is potentially important. However, the authors did not compare the effects of D-2-HG and L-2-HG on not all the aspects of DC functions. Thus, the authors should examine the effects of both D-2-HG and L-2-HG on all the aspects of DC functions. Moreover, the authors did examine only the uptake of D-2-HG at 24 hrs but did not examine L-2-HG uptake. Considering that the authors examined the effects of HGs on DC functions at the time points earlier than 24 hrs, they should determine D-2-HG and L-2-DG uptake in a time sequential manner.

Specific comments

#1. The authors should examine the effects of either D-2-HG or L-2-HG on DC phenotypes and functions in the absence of LPS stimulation.

#2. The authors should provide the rationale to choose the used concentrations of D-2-HG and L-2-DG in the experiments.

#3. Figure 2C, 3D, 4B, and 4C. The variations were too large as in vitro experiments, raising a serious question of the validity as a whole.

Author Response

The authors should examine the effects of either D-2-HG or L-2-HG on DC phenotypes and functions in the absence of LPS stimulation.-We included a new Figure (Figure 1e) where we incubated DCs without LPS in the presence of 2-HG and determined surface markers. In addition, we determined the capacity of iDCs for antigen presentation  (without LPS stimulation) in a mixed lymphocyte reaction (Figure 2f). 

#2. The authors should provide the rationale to choose the used concentrations of D-2-HG and L-2-DG in the experiments.-We included two new figures (2a/b) where we titrated the HG concentrations. 

#3. Figure 2C, 3D, 4B, and 4C. The variations were too large as in vitro experiments, raising a serious question of the validity as a whole.-As we work with primary human cells, each dot represents another donor which explains the high variation  in the results. 

Round 2

Reviewer 2 Report

The authors modified the manuscript fully in response to the comments.